# Clinical and Radiological Outcomes of a Comparative Study of Anterior Cervical Decompression and Fusion with Partial Pediculotomy, Partial Vertebrotomy (PPPV) Posterior Endoscopic Cervical Decompression (PECD) for Cervical Foraminal Pathology

**DOI:** 10.3390/medicina59071222

**Published:** 2023-06-29

**Authors:** Hyeun Sung Kim, Pang Hung Wu, Brian Zhao Jie Chin, Il Tae Jang

**Affiliations:** 1Harrison Spinartus Hospital, Chungdam, Seoul 06084, Republic of Korea; 2Achieve Spine and Orthopaedic Centre, Mount Elizabeth Hospital, Singapore 228510, Singapore; 3Orthopaedic Surgery, Jurong Health Campus, National University Health System, 1 Jurong East Street 21, Singapore 609606, Singapore; 4Spines Surgery, Nanoori Gangnam Hospital, Seoul 06048, Republic of Korea; nanooriresearch@gmail.com

**Keywords:** cervical radiculopathy, endoscopic spine surgery, posterior endoscopic cervical discectomy, posterior endoscopic cervical foraminotomy, cervical spine, cervical endoscopy, cervical alignment

## Abstract

*Background and Objectives*: The purpose was to compaSre medium-term clinical and radiological outcomes of Partial Pediculotomy, Partial Vertebrotomy (PPPV) Posterior Endoscopic Cervical Decompression (PECD) surgery versus Anterior Cervical Discectomy and Fusion (ACDF) for patients with cervical disc herniations and foraminal pathologies. *Materials and Methods*: A prospective registry of patients who had undergone either PPPV PECD surgery or ACDF surgery for cervical disc herniation or foraminal pathologies under a single fellowship-trained spine surgeon was performed. The baseline characteristics and operative details including complications were recorded for all included patients. The clinical outcomes evaluated include VAS, MJOA, motor score, and NDI and MacNab’s score. The radiological parameters in neutral-measured facet length, facet area, disc height, C2–C7 angle, neck tilt angle, T1 slope and thoracic inlet angle were also evaluated. *Results*: A total of 55 patients (29 PPPV PECD, 26 ACDF) were included, with mean follow-up periods of 21.9 and 32.3 months, respectively. Each cohort was noted to have a single case of surgical complication. Statistically significant changes of facet area (49.05 ± 14.50%) and facet length (52.71 ± 15.11%) were noted in the PPPV PECD group. At neutral alignment of the neck on a lateral X-ray, compared to ACDF, PPPV PECD had a statistically significant change in neck tilt angle (−11.68 ± 17.35°) and T1 slope angle (−11.69 ± 19.58°). Whilst both PPPV PECD and ACDF had significant improvements in VAS, MJOA and NDI postoperatively, PPPV PECD was found to be superior across all above scores at various follow-up timepoints compared to its ACDF counterparts. *Conclusions*: PPPV PECD surgery achieved a satisfactory radiological correction of neck alignment and significantly improved clinical outcomes at medium-term follow-up for our cohort of patients, highlighting its feasibility in treating patients with cervical disc herniations and foraminal pathologies.

## 1. Introduction

Cervical radiculopathy, described as cervical nerve root irritation and dysfunction secondary to compression in the cervical foraminal region [1,2], is a common cause of significant neck disability with debilitating consequences [1,2]. Despite trials of conservative treatments with physiotherapy and symptomatic medications, symptomatic relief may take up to 2 to 3 years, with 17% of the patients reported to suffer disabling symptoms at long-term follow-up [3]. Anterior cervical discectomy and fusion and artificial disc replacement are popular surgical options offered to patients with cervical radiculopathy who have failed initial conservative treatment [4]. While these anterior cervical surgical treatment options are effective, they are associated with anterior-approach-related complications, such as dysphagia and a hoarseness of the voice, and implant-related complications such as adjacent-segment degeneration and pseudoarthrosis of the operative level—with an overall complication rate of anterior cervical surgery ranging from 13.2 to 19.3% [5]. Posterior cervical foraminotomy (PCF) has gained popularity as an alternative to anterior cervical discectomy and fusion, sparing patients from anterior-approach-related complications and problems associated with fusion and surgical instrumentations while offering equally good clinical results and the added advantage of motion preservation [6,7,8]. Traditionally, PCF is performed as an open or mini-open procedure, with associated complications of posterior neck wound pain, infection of the open posterior wound, and an over resection of the facet joint leading to instability. Moreover, the removal of the cervical disc via a posterior approach is technically more demanding compared to anterior approaches, often warranting conversion to open approaches [9]. With the evolution of endoscopic techniques and advancements in surgical technology, posterior endoscopic cervical decompression (PECD) has been gaining traction as a minimally invasive technique for cervical foraminal decompression and discectomy [10]. PECD has also been shown to improve perioperative pain, decrease infection rate and effectively relieve cervical radiculopathy symptoms [10,11,12]. To mitigate risks of neurologic complications (associated with spinal cord and exiting nerve root retraction) or inadequate decompression, Kim and Wu et al. described a safe approach to the cervical intervertebral disc via partial pediculotomy and partial vertebrotomy posterior endoscopic cervical decompression (PPPV PECD), permitting more subneural space by drilling the pedicle and vertebral body to create a working space to gain access to the disc [13,14]. Hitherto, there is paucity in the literature describing radiological changes in the postoperative CT scan and XR alignment changes of patients who underwent PPPV PECD. In this study, we followed up prospectively with a cohort of patients who underwent PPPV PECD to evaluate the effect of facet joint dimensions, disc height, cervical alignment and clinical outcomes after PPPV PECD. Additionally, the cohort was compared with a separate cohort of patient who underwent ACDF surgery to better understand potential differences in clinical outcomes. 

## 2. Materials and Methods

### 2.1. Patients’ Demographics

Institutional review board (IRB) approval was obtained prior study commencement (IRB 2022-009). Patients enrolled included those with clinical presentation of unilateral intractable cervical radiculopathy who failed to achieve symptomatic control with at least 6 weeks of conservative treatment comprising analgesia (NSAIDs) and physiotherapy and magnetic resonance imaging finding of cervical nerve root compression from cervical foraminal stenosis contributed from a prolapsed or bulging lateral degenerative cervical disc with hypertrophy in uncovertebral joint and/or facet joint. For patients with prolapsed cervical intervertebral disc, we included patients who had 2/3 of cervical intervertebral disc material lateral to the lateral margin of the thecal sac on axial cut. Patients with calcified central disc, cervical spinal instability, more than 10 degrees of cervical kyphosis, cervical myelopathy, and predominant axial neck pain were excluded from the study. 

Enrolment period was from April 2018 to September 2020, with clinical follow-up of at least 1 year postoperatively. Included patients consented for single- or double-level Uniportal Full Endoscopic Partial Pediculotomy and Partial Vertebrotomy Posterior Endoscopic Cervical Decompression (PPPV PECD). This cohort was compared to 26 patients who previously underwent ACDF surgery from January 2016 to December 2018, with similar inclusion criteria as above. 

### 2.2. Surgical Technique—Partial Pediculotomy and Partial Vertebrotomy Posterior Endoscopic Cervical Decompression 

All procedures were performed under general anesthesia [13,14]. The patient is first positioned prone on rolled gel cushions with shoulders strapped down and neck flexed in a slight reverse Trendelenburg position [15]. A single dose of intravenous antibiotics is administered to all patients. PPPV PECD was performed by first docking at the cervical facet “V” point located at the lateral margin of interlaminar space and medial border of junction of facet joint. A transverse 8 mm incision was made at the “V” point and serial obturators were inserted to dock a uniportal endoscope under fluoroscopic guidance at the appropriate vertebral level. Localization was performed after identifying bony landmarks following soft tissue dissection with endoscopic forceps and radiofrequency ablation. Bony decompression was then performed by drilling on medial aspect of the lateral mass and facet joint, with typically around a 3 mm diameter of bone removed from the inferolateral aspect of upper lamina, followed by about 3 mm of the inferomedial portion of the inferior articular facet from the lamina–facet border (“V”point) to gain access to the nerve root (Figure 1A) [16]. Subsequently, the upper medial quadrant of the caudal vertebra pedicle was drilled (partial pediculotomy)—with around 3 mm depth of caudal vertebra subneural indented space (partial vertebrotomy) for easier access to the disc (Figure 1B–D). This added PPPV technique allows retrieval of disc material and uncus decompression without cord retraction. Upon completion of decompression, the exiting nerve root and lateral third of spinal cord were assessed under endoscopic visualization to ensure they were free of compression (Figure 1E,F). 

### 2.3. Surgical Technique—Anterior Cervical Discectomy and Fusion (ACDF)

All routine ACDF surgeries were performed in supine position with bolster under the neck. The Smith-Robinson approach was used to gain access to the level of interest. Discectomy was then performed with resection of the posterior longitudinal ligament and insertion of PEEK cage and plate to complete the ACDF procedure. Subsequently, the neck wound is closed in layers with a drain inserted.

### 2.4. Clinical and Radiological Outcomes Evaluation

All patients were followed up at postoperative 1 week, 3 months, 6 months, and final follow-up periods. Clinical data and functional scores such as the Visual Analogue Scale (VAS) [17], Neck Disability Index (NDI) [18] and MacNab’s Criteria [19] were collected at each visit. Data from clinical records were analyzed by 2 fellowship-trained spine surgeons who were not involved in the care of study subjects. Workstation software (Infinitt, Inc., Seoul, Republic of Korea) was used for measurement of radiological variables, conducted by 2 fellowship-trained spine surgeons who were not involved in surgery of the study subjects. Preoperative Computed Tomography (CT), plain flexion, extension and anteroposterior and neutral X-ray films were obtained for all included patients. Patients with motion >3 mm at the operative segment on dynamic roentgenogram imaging were excluded for instability [20]. The 3D reconstructed Computed Tomographic cervical spine images were obtained preoperatively and at one-year follow-up—with measurement of the CT facet length and 3D reconstructed CT facet. Xray measurements were made at preoperative and postoperative one-year periods, with recorded disc height, C2–C7 angle, neck tilt angle, T1 slope and thoracic inlet angle (Figure 2 and Figure 3).

### 2.5. Statistical Analysis

All data were analyzed using IBM SPSS Statistics Software Application 26.0 version (IBM Corporation, Armonk, NY, USA). Continuous variables were presented as means with standard deviations (SD), whereas discontinuous variables were expressed as percentages. Statistical analyses comparing clinical and radiological results pre and postoperatively were performed using paired *t*-test, with *p* < 0.05 defined as statistical significance. 

## 3. Results

### 3.1. Baseline Demographics and Operative Data

A total of 55 patients (29 PPPV PECD, 26 ACDF) with a mean age of 52.2 and 57.1 years respectively were enrolled in the present study. Of these, 24 patients (3 of C4/5, 16 of C5/6, 5 of C6/7) and 5 patients (1 each of C3–C5, C4–C6, 3 of C5–C7) underwent single or double-level PPPV PECF, respectively. The mean follow-up duration of included patients who underwent PPPV PECD and ACDF was 21.9 (14–35) and 32.31 (12–115) months, respectively. The mean operative duration per level was 58.62 ± 15.58 and 123.46 ± 44.02 min, respectively, for PPPV PECD and ACDF (*p* < 0.05). There was no statistical difference in outcomes affected by sex and age (Table 1). 

### 3.2. Clinical Outcomes

The complication rate of PPPV PECD in our cohort was 3%, with a single case of postoperative C5 palsy—subsequently resolved within 1 year without neurological sequelae. One case of incomplete decompression was noted in the ACDF cohort. Statistically significant improvements in Visual Analog Scale (VAS), Modified Japanese Orthopaedic Association (MJOA), Neck Disability Index (NDI) and motor arm power were observed across all follow-up timings (postop 1 week, 3 months, final follow-up; taken at patient’s last outpatient appointment) in comparison with preoperative scores for both PPPV PECD and ACDF cohorts (Table 2). Patients who underwent PPPV PECD had significantly better MJOA scores at 2 weeks and 3 months postoperatively; VAS at 3 months postoperatively; and NDI across all follow-up timepoints when compared to the ACDF cohort. Of note, no statistically appreciable differences in upper limb motor score were found when comparing patients who underwent PPPV PECD or ACDF across all follow-up timepoints. 

### 3.3. Radiological Outcomes

Analysis of postoperative 3D CT scans for the PPPV PECD cohort revealed a statistically significantly increased facet area (49.05 ± 14.50%) and a reduced facet length (52.71 ± 15.11%) when compared with preoperative scans. At postoperative one-year, there was statistically significant change in C2–C7 angle (4.16 ± 8.71°), neck tilt angle (−11.68 ± 17.35°) and T1 slope (−11.69 ± 19.58°) degrees, respectively. No statistically significant change in the disc height and the thoracic inlet angle were found (Table 3). The PPPV PECD cohort had significantly lower measured neck tilt angle and T1 slope angle postoperatively when compared to the ACDF cohort. 

## 4. Discussion

Anterior cervical discectomy and fusion and anterior cervical disc replacement are popular and effective surgeries offered to patients with cervical radiculopathy; however, they are associated with anterior-approach-related complications, risk of adjacent segment degeneration, and implant-related complications [5,21]. Posterior endoscopic cervical decompression is an emerging technique which has been described to decrease approach-related complications and mitigate postoperative morbidities by minimizing soft tissue damage [22,23,24]. This technique, however, is not without drawbacks—whereby access to disc space can be challenging due to limited retraction of the cord. The PPPV PECD technique then was devised to address this concern by creating a subneural space to safely place a working retractor, thus allowing a safe retrieval of the cervical disc with endoscopic forceps [14].

Various authors had reported positive clinical outcomes in their retrospective PECD studies [6,13,14,25]. In the present study, our patients also experienced good improvement in motor power, VAS, MJOA and NDI. There is a low complication rate of 3% (involving a case of C5 palsy) in our series, which is grossly similar to available studies on PPPV PECD [5,12,26,27]. The authors felt that PECD for the level of C4/5 has a correlation with complications of C5 palsy and advocate for more studies to elucidate the effect of C5 palsy. Despite concerns of potential cervical spinal instability and kyphosis following PECD, several existing studies have demonstrated that PECD effectively preserved the disc height and the range of motion of the cervical spine as well as maintained cervical stability [13,28,29,30]. Similarly, our study reflected no significant change in disc height, with no cases of cervical instability in our cohort of patients despite a statistically significant facet change of around 50% in axial and CT 3D reconstruction cuts. 

There is a paucity of studies evaluating cervical spinal alignment in cervical endoscopic spine surgery, of which the majority are based on anterior approaches [31,32,33,34]. In our study, we focused on evaluating cervical spinal alignment preoperatively and postoperatively at one-year follow-up to investigate alignment changes post PECD. Our study reflected a statistically significant increase in lordosis in C2–7 (4.16 ± 8.71°), a decrease in neck tilt angle (−11.68 ± 17.35°) and T1 slope (−11.69 ± 19.58°) at a minimally one-year follow-up compared to the preoperative state. This change in alignment is postulated to be reflective of a reduction in muscle spasm and cervical neck pain leading to an improved cervical alignment [35]. Similar results were found in existing PECD studies [33,34]. More studies are needed to evaluate the cervical spine alignment changes post PPPV PECD as we consider the indications for PPPV PECD in comparison to ACDF and ADR for cervical radiculopathy. 

## 5. Limitations of the Study

There were inherent limitations in this retrospective cohort study. Preoperative medical comorbidities were not collected which might introduce confounders in the study. The cohort’s small sample size and medium term follow-up period were inherent limitations of this study. The authors suggest longer follow-up studies of a greater size to evaluate the effectiveness of PPPV PECD and make comparisons with ACDF or ADR. 

## 6. Conclusions

PPPV PECD is an effective alternative treatment for cervical radiculopathy and has significantly improved clinical outcomes and cervical spinal alignment in the medium term for the included cohort of patients. 

## Figures and Tables

**Figure 1 medicina-59-01222-f001:**
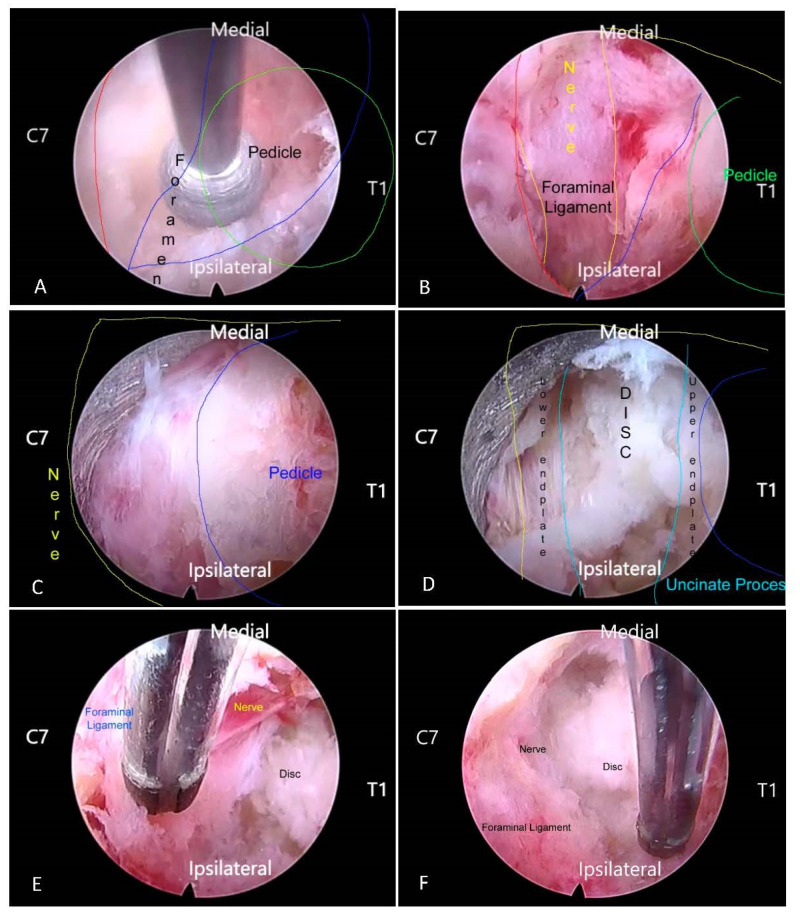
Partial Pediculotomy, Partial Vertebrotomy, Posterior Endoscopic Cervical Decompression PPPV PECD Technique. (**A**) Endoscopic drilling of left C7/T1 laminofacect junction (Vpoint). (**B**) Exposure of foraminal ligament and exiting nerve root. (**C**) Endoscopic drilling of the pedicle (partial pediculotomy). (**D**) After endoscopic drilling of the vertebral(partial vertebrotomy), disc is exposed. (**E**) Exiting nerve root hemostasis. (**F**) Disc space exposed.

**Figure 2 medicina-59-01222-f002:**
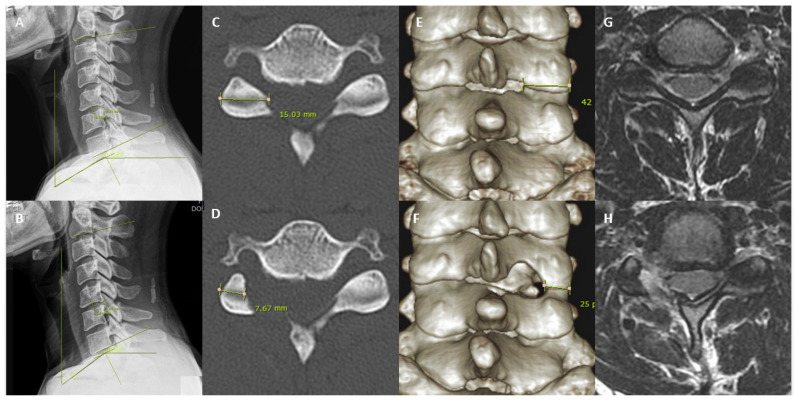
Pre and postoperative one-year XR, CT and MRI of a patient who underwent partial pediculotomy and partial vertebrotomy posterior endoscopic cervical decompression of right cervical five cervical six. (**A**) Preoperative measure at neutral of C2–C7 angle, neck tilt angle, disc height, T1 slope and thoracic inlet angle was compared with postoperative one year (**B**). (**C**) Preoperative CT axial cut measured the facet length at the foraminal level compared to (**D**) postoperative CT axial cut facet length. (**E**) Preoperative CT 3D reconstruction coronal cut demonstrates intact (**F**) postoperative one-year CT 3D reconstruction coronal; (**G**) Preoperative MRI axial cut (**H**) postoperative one-year MRI axial cut.

**Figure 3 medicina-59-01222-f003:**
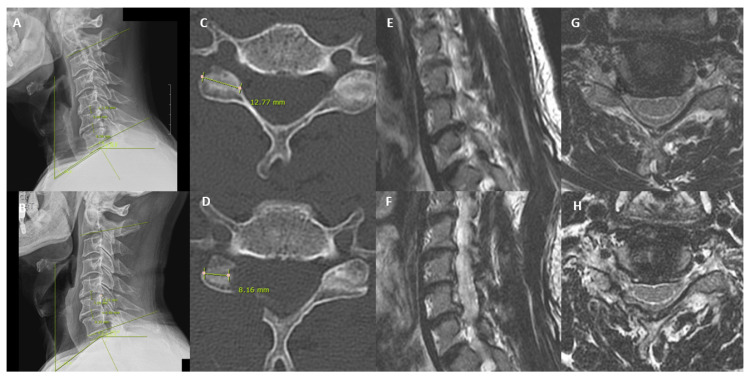
Pre and postoperative one year XR, CT and MRI of a patient who underwent partial pediculotomy and partial vertebrotomy posterior endoscopic cervical decompression of right C5/6 and C6/7. (**A**) Preoperative measure at neutral of C2–C7 angle, neck tilt angle, disc height, T1 slope and thoracic inlet angle was compared with postoperative one year (**B**). (**C**) Preoperative CT axial cut measured the facet length at the foraminal level compared to (**D**) postoperative CT axial cut facet length. (**E**) Preoperative MRI sagittal cut demonstrated right C5/6 and C6/7 foraminal stenosis compared to (**F**) postoperative one-year sagittal cut demonstrated the decompressed right C5/6 and C6/7. (**G**) Preoperative MRI axial cut of C5/6 demonstrates foraminal stenosis which was relieved in postoperative MRI axial cut of the same level in (**H**).

**Table 1 medicina-59-01222-t001:** Baseline and operative characteristics.

	PPPV PECD	ACDF
Total cases	29	26
Age (years)	52.21 (24–72)	57.12 (45–80)
*n* of males (%)	18 (62)	17 (65)
Mean follow-up time (months)	21.9 (14–35)	32.31 (12–115)
Spinal Levels Operated (%)		
1	5 (17.2)	-
2	24 (82.8)	-
Surgery Duration Per Level Operated (Mins)	58.62 ± 15.58	123.46 ± 44.02
Patients requiring blood transfusion	0	0
Intraoperative Complications (%)	1 (3.4)	1 (3.8)
Conversion to open surgery (*%*)	0	-
Relook surgeries (*%*)	0	0

**Table 2 medicina-59-01222-t002:** Clinical outcomes of posterior partial pediculotomy partial vertebrotomy posterior endoscopic cervical decompression versus anterior cervical decompression and fusion.

	PPPV PECD	*p*-Value (Comparison to Preop)	ACDF	*p*-Value (Comparison to Preop)	*p*-Value (PPPV PECD vs. ACDF)
**Motor score (Upper Limb)**					
Preoperative	4.07 (0–5)	-	4.27 (0–5)	-	-
1-week postoperative	4.69 (2–5)	0.006 *	4.69 (2–5)	0.002 *	0.307
3-months postoperative	4.79 (2–5)	<0.001 *	4.85 (3–5)	0.003 *	0.544
Final follow-up	4.97 (4–5)	<0.001 *	4.96 (4–5)	0.005 *	0.498
**MJOA score**					
Preoperative	16.34 (13–17)	-	16.5 (14–17)	-	-
1-week postoperative	16.76 (14–18)	0.046 *	16.77 (15–17)	0.006 *	0.378
3-months postoperative	17.34 (16–18)	<0.001 *	16.89 (16–17)	0.005 *	0.001 *
Final follow-up	17.66 (17–18)	<0.001 *	16.96 (16–17)	0.005 *	<0.001 *
**Visual analogue score**					
Preoperative	7.31 (3–10)	-	7.04 (5–10)	-	-
1-week postoperative	3.07 (2–4)	<0.001 *	3.35 (2–4)	<0.001 *	0.155
3-months postoperative	2.03 (1–3)	<0.001 *	2.42 (1–4)	<0.001 *	0.096
Final follow-up	1.38 (1–2)	<0.001 *	2.23 (1–3)	<0.001 *	0.013 *
**Neck disability index**					
Preoperative	73.86 (54–88)	-	67.31 (48–88)	-	-
1-week postoperative	30.55 (24–46)	<0.001 *	32.23 (24–42)	<0.001 *	0.004 *
3-months postoperative	24.14 (20–32)	<0.001 *	26.54 (18–42)	<0.001 *	0.002 *
Final follow-up	20.62 (16–24)	<0.001 *	25 (16–38)	<0.001 *	<0.001 *

*: statistically significant.

**Table 3 medicina-59-01222-t003:** Radiological outcomes of posterior partial pediculotomy partial vertebrotomy posterior endoscopic cervical decompression versus anterior cervical decompression and fusion.

	PPPV PECD	*p*-Value (Comparison to Preop)	ACDF	*p*-Value (Comparison to Preop)	*p*-Value (PPPV PECD vs. ACDF)
**C2–C7 angle at neutral (°)**					
Preoperative	5.08 ± 9.20	-	8.54 ± 11.42	-	-
Postoperative	9.24 ± 9.61	0.049 *	8.85 ± 9.67	0.833	0.085
**Neck tilt angle (°)**					
Preoperative	55.05 ± 7.22	-	52 ± 9.71	-	-
Postoperative	43.37 ± 17.79	<0.001 *	53.35 ± 8.05	0.271	<0.001 *
**T1 slope angle (°)**					
Preoperative	32.23 ± 17.60	-	23.92 ± 6.71	-	-
Postoperative	20.54 ± 6.50	<0.001 *	25.54 ± 7.02	0.148	0.002 *
**Thoracic inlet angle (°)**					
Preoperative	75.65 ± 7.38	-	75.92 ± 10.86	-	-
Postoperative	75.90 ± 6.83	0.447	78.89 ± 9.05	0.079	0.111
**Facet length (mm)**					
Preoperative	15 ± 1.79	-	-	-	-
Postoperative	7.89 ± 2.4	<0.001 *	-	-	-
**Facet area (mm^3^)**					
Preoperative	44.21 ± 6.78	-	-	-	-
Postoperative	21.55 ± 7.27	<0.001 *	-	-	-
**Disc height (mm)**					
Preoperative	6.41 ± 1.39	-	-	-	-
Postoperative	6.38 ± 1.41	0.354	-	-	-

*: statistically significant.

## Data Availability

Data available upon contact with authors.

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
