# Peer review of "Clinical and Radiological Outcomes of a Comparative Study of Anterior Cervical Decompression and Fusion with Partial Pediculotomy, Partial Vertebrotomy (PPPV) Posterior Endoscopic Cervical Decompression (PECD) for Cervical Foraminal Pathology"

_medicina, 2023, doi:10.3390/medicina59071222_

Round 1

Reviewer 1 Report (Previous Reviewer 2)

I cannot find any images specific to this surgical procedure. It is difficult to distinguish it from a simple posterior intervertebral foraminotomy.

If you want to show the effectiveness of this procedure, please show images that demonstrate the actual procedure, e.g., endoscopic views of a vertebroplasty or pediculectomy being performed.

I think this paper is worthless without more emphasis on the originality of the technique.

You should also consider the inferior postoperative results of cervical disc herniation. Usually, ACDF has better results.

Furthermore, although the imaging evaluation shows that the cervical alignment has improved, there is no consideration of a smaller neck tilt or T1 slope, or a larger cervical kyphosis. Without considering that, the value of this paper will not increase.

Author Response

We thank the reviewer for perusing our work and his/her valuable input. We have now added additional images to better portray the surgical procedure.

Based on our results, PPPV PCED results in smaller postop neck tilt angle and T1 slope angle compared to ACDF. Postoperative cervical kyphosis was comparable in both treatment arms.

We would like to clarify regarding the reviewer's question, specifically: "consider the inferior postoperative results of cervical disc herniation", does the reviewer mean for us to compare ACDF versus PPPV PCED outcomes for treatment of cervical disc herniation?

Reviewer 2 Report (Previous Reviewer 3)

Reviewer’s Comment

In line 98: Upon completion of decompression, the exiting nerve root and lateral third of spinal cord  were assessed (by what means?) to[A1] [A2]  ensure they were free of compression.

In line 112: Typo errors. Workstation software was used for measurement[A3]  of radiological variables[A4] , conductedby 2 fellowship trained spine surgeons who were not involved in surgery[A5]  of the study subjects.

In line 159: Neck Disability Index (NDI) and motor arm power was observed across all follow up timings (postop 1 week, 3 months, final[A6]  follow-up – define what is the final follow-up).

Table 2. Clinical Outcomes (PPPV PCED vs ACDF). Under the outcome motor score (upper limb), the authors’ stated Final follow-up, they should specify what this final follow-up means.

The following outcome measures: Visual Analog Scale (VAS), Modified Japanese Orthopaedic Association (MJOA), Neck Disability Index (NDI), only had 3 months as the final follow-up. However, from lines 157-160, the authors’ stated “Statistically significant improvements in Visual Analog Scale (VAS), Modified Japanese Orthopaedic Association (MJOA), Neck Disability Index 158 (NDI) and motor arm power was observed across all follow up timings (postop 1 week, 3 months, final follow-up) in comparison with preoperative scores for both PPPV PECD and ACDF cohorts (Table 2)’. See highlighted phrase in yellow colour.

Under the discussion section, lines 206-207, the authors stated: “Our study reflected statistically significant increase in lordosis in C2-7 (4.16 ± 8.71°), decrease in neck tilt angle (-11.68 ± 17.35°) and T1 slope (-11.69 ± 19.58°) at one year follow up compared to preoperative state”. However, in the abstract section on lines 14-15, they stated “A total of 55 patients (29 PPPV PCED, 26 ACDF) were included, with mean follow-up periods of 21.9 and 32.3 months respectively

Summary:

1.      This is good research with valuable findings for patients suffering from medium and long-term cervical radiculopathy that have failed conservative therapy, however, the work needs proofreading by an English Language Professional or by using simple soft wear as “Grammarly”. I am not advertising this soft wear by any means.

2.      They need to be very clear regarding the follow-up periods of this study. There are ambiguities regarding different statements regarding the end-point follow-up, it is 3 months, 12 months or more?

3.      Finally, once these concerns are addressed the work can be considered for publication. Thank you

1.      This is good research with valuable findings for patients suffering from medium and long-term cervical radiculopathy that have failed conservative therapy however, the work needs proofreading by an English Language Professional or by using simple soft wear as “Grammarly”. I am not advertising this soft wear by any means.

Author Response

1. Reply to reviewer: We thank the reviewer for clarifying this point – the exiting nerve root and lateral third of spinal cord were assessed via endoscopic visualization to ensure they were free of compression. This has now been added into the manuscript.

2. Reply to reviewer: Thank you for pointing out the typo errors – we have promptly made the appropriate changes.

3. Reply to reviewer: With regards to final follow-up, there were patients who had outcome scores recorded at their last outpatient appointment at our tertiary hospital. These scores were minimally recorded at 12 months postoperatively for ACDF and 14 months postoperatively for PPPV PCED. We have defined this in our recent edits of the current manuscript.

4. Reply to reviewer: Clarified as above.

5. Reply to reviewer: We thank the reviewer for pointing out the errors above – the tables have been edited to reflect the correct follow-up values at postop 1 week, 3 months, and final follow-up.

6. Reply to reviewer: We thank the reviewer for clarifying – the above sentence have been edited to include minimally one year follow-up.

Round 2

Reviewer 1 Report (Previous Reviewer 2)

The presentation of appropriate key images made it easier to understand.

This manuscript is a resubmission of an earlier submission. The following is a list of the peer review reports and author responses from that submission.

Round 1

Reviewer 1 Report

Comments to the Author I thank you for the opportunity to review your paper. The authors followed up with a cohort of patients who underwent PPPV PECD prospectively to evaluate the effect of facet joint dimensions, disc height, cervical alignment, and clinical outcomes after PPPV PECD. A retrospective cohort study of 29 patients with cervical endoscopic decompression was performed. >From the results, the authors concluded that PPPV PECD significantly improves clinical outcomes and neck alignment for our cohort of patients in the medium term. The authors demonstrated that “There were inherent limitations in this retrospective cohort study. There was no control group to compare in this study as the authors did not perform ACDF or open posterior cervical foraminotomy during the study period for isolated cervical radiculopathy. Pre-operative medical comorbidities were not collected which might introduce confounders in the study. The cohort’s small sample size and medium term follow up period were inherent limitations of this study, longer follow up study to evaluate the treatment effectiveness and reoperation rate would improve the quality of the study.” I agree with this sentence. The study had only 29 cases and no controls. This cannot determine the usefulness of this study. In order to make this paper worthwhile, I recommend increasing the number of cases and comparing them with other surgical procedures. In the current form of the paper, I cannot agree to accept it.

Reviewer 2 Report

It is understood that this surgical technique improved the clinical data and cervical alignment.

I consider it to be an excellent technique. It would be easier to understand if you could illustrate the technique a little more and show the key intraoperative fields of view.

The results are presented in a table, but please consider creating an easy-to-understand table.

Reviewer 3 Report

- What is the duration of the outcome measures?

- What is the final follow-up period?

- Authors to reconsider their referencing - please refer to the comments

- No mention of sample size calculation

- The tables are no too to follow

- No mention of adverse risk factors in the results section but is was discussed in the discussion section

Reviewer 4 Report

Major problems: 

(1)As for PECD, the partial pediculotomy and vertebrotomy are only routinely performed in some patients. Defining the patient selection criteria is essential while evaluating a new surgical technique. For example, the surgeon should consider the herniated disc's location and degree of migration. Though it was a retrospective study, the readers should know the ideal candidate for the PPPV technique.  

(2)In the discussion section, the authors assumed that the alignment improvement resulted from the improvement of muscle spasms and neck pain. However, there were no results supported their assumption. The authors claimed the benefits of PECD with PPPV, but the thesis lacked scientific evidence. Besides, the discussion section needed to be deeper, with new findings and value regarding the PECD. The authors also mentioned that similar results are found in other PECD studies. Then, why should we use the PPPV technique if the outcomes were similar between the current study and previous ones? Therefore, the authors should propose more data and have a deep discussion to support their conclusion. 

Minor problems: 

(1)Line 159-160, the author stated that there is a "significant change in facet area in postoperative 3D CT scan as compared to preoperative CT scans (49.05±14.50%) increase and facet length (52.71±15.11%) decrease respectively." I saw both facet area and length decrease in the postoperative CT. Is there any error in the sentence? Please clarify it. 

(2)Table 1 was overlong and contained too much outcome data. Please report patient demographic data in table 1 only. 

(3)In the other tables, there were too many rows in a table. It takes much work for readers to interpret all data in a single column. Please revise the table by adding more columns to make it an easy read. 

(4)There were many minor grammatical errors in the article. I recommend that the authors seek professional English proofreading services. v